Genome-wide identification and expression analysis of the WRKY gene family in Rhododendron henanense subsp. lingbaoense

Guo Xiangmeng 23511243@qq.com
Yan Xinyu
Li Yonghui liyonghui@lynu.edu.cn
School of Life Sciences, Luoyang Normal University , Luoyang, Henan , China
Singh Kashmir
Electronic publication date: 2024 May 29
Publication date: 2024
Volume: 12
Electronic Location ID: e17435
Received 2024 Jan 24; Accepted 2024 Apr 30
Copyright: © 2024 Guo et al.
Copyright year: 2024
Copyright holder: Guo et al.
License: This is an open access article distributed under the terms of the Creative Commons Attribution License, which permits unrestricted use, distribution, reproduction and adaptation in any medium and for any purpose provided that it is properly attributed. For attribution, the original author(s), title, publication source (PeerJ) and either DOI or URL of the article must be cited.
License URL: https://creativecommons.org/licenses/by/4.0/

Keywords: Rhododendron henanense subsp. Lingbaoense (Rhl), WRKY, Bioinformatics, Abiotic stress, Expression patterns

Funding: Key Scientific Research Projects of Higher Education Institutions in Henan Province 24A180017 Innovative Training Program for College Students in Henan Province 202310482045 This work was supported by the Key Scientific Research Projects of Higher Education Institutions in Henan Province (24A180017) and the Innovative Training Program for College Students in Henan Province (202310482045). The funders had no role in study design, data collection and analysis, decision to publish, or preparation of the manuscript.

==============================
Background

This work explored the characteristics of the WRKY transcription factor family in Rhododendron henanense subsp. lingbaoense (Rhl) and the expression patterns of these genes under abiotic stress by conducting bioinformatics and expression analyses.

Methods

RhlWRKY genes were identified from a gene library of Rhl. Various aspects of these genes were analyzed, including genetic structures, conserved sequences, physicochemical properties, cis-acting elements, and chromosomal location. RNA-seq was employed to analyze gene expression in five different tissues of Rhl: roots, stems, leaves, flowers, and hypocotyls. Additionally, qRT-PCR was used to detect changes in the expression of five RhlWRKY genes under abiotic stress.

Result

A total of 65 RhlWRKY genes were identified and categorized into three subfamilies based on their structural characteristics: Groups I, II, and III. Group II was further divided into five subtribes, with shared similar genetic structures and conserved motifs among members of the same subtribe. The physicochemical properties of these proteins varied, but the proteins are generally predicted to be hydrophilic. Most proteins are predicted to be in the cell nucleus, and distributed across 12 chromosomes. A total of 84 cis-acting elements were discovered, with many related to responses to biotic stress. Among the identified RhlWRKY genes, there were eight tandem duplicates and 97 segmental duplicates. The majority of duplicate gene pairs exhibited Ka/Ks values <1, indicating purification under environmental pressure. GO annotation analysis indicated that WRKY genes regulate biological processes and participate in a variety of molecular functions. Transcriptome data revealed varying expression levels of 66.15% of WRKY family genes in all five tissue types (roots, stems, leaves, flowers, and hypocotyls). Five RhlWRKY genes were selected for further characterization and there were changes in expression levels for these genes in response to various stresses.

Conclusion

The analysis identified 65 RhlWRKY genes, among which the expression of WRKY_42 and WRKY_17 were mainly modulated by the drought and MeJA, and WRKY_19 was regulated by the low-temperature and high-salinity conditions. This insight into the potential functions of certain genes contributes to understanding the growth regulatory capabilities of Rhl.

Introduction

Rhododendron henanense subsp. lingbaoense (hereinafter referred to as Rhl) is a subspecies of Rhododendron henanense native to the western part of Henan Province, concentrated within the Xiaoqinling National Nature Reserve (Yue, Han & Zhang, 2019). Rhl is known for its vibrant and clustered blossoms, making it a highly valued ornamental plant. However, its habitat is currently in decline, resulting in a rapid decrease in plant numbers, putting this subspecies in a precarious state. As a unique species, Rhl is a valuable plant to study with significant natural heritage value and the added importance of being a repository of genetic resources (Zhou et al., 2019). In recent years, many scholars, both domestically and internationally, have shown considerable interest in studying the medicinal ingredients of different Rhododendron species, such as the inhibition of liver cancer cell activity by lupeol and uvaol from Rhododendron micranthum (Chang, 2011) and the treatment of vascular inflammation by hyperoside from Rhododendron brachycarpum G. Don (Ku et al., 2015). Overall, components of Rhododendron species hold potential for various applications, including ecological conservation, aesthetics, medicinal uses, and scientific research (Li et al., 2018; Liang et al., 2016) Plants continuously face various stressors, during their growth potentially limiting their growth and development (Chen et al., 2012). At the molecular level, the induction of stress resistance genes helps plants adapt to unfavorable environmental conditions (Matsui et al., 2008). Among these genes, WRKY transcription factors are part of a large family of plant regulatory proteins. The most distinctive feature of WRKY transcription factors is a DNA-binding domain composed of approximately 60 amino acids, with a highly conserved heptapeptide sequence WRKYGQK at the N-terminus (Jiang et al., 2015), which gives this family its name. WRKY transcription factors are classified into three major categories, WRKY I, II, and III, based on differences in their conserved domains. WRKY I possesses two WRKY conserved domains with a C2H2 zinc finger, WRKY II contains only one WRKY conserved domain with a C2H2 zinc finger, and WRKY III has one WRKY conserved domain with a C2HC zinc finger (Vives-Peris et al., 2018). The regulation of gene expression by WRKY transcription factors primarily occurs through binding to specific cis-regulatory elements called W-box (TTGACC) elements. This binding can activate or suppress the transcription of downstream target genes and is also subject to regulation by upstream master regulatory proteins (Huang et al., 2019). WRKY transcription factors play crucial roles in plant growth, development, and responses to abiotic stresses. For instance, WRKY transcripts were expressed in elongating fiber ovules in cotton three days after flowering, suggesting involvement of these transcription factors in fiber development (Wang et al., 2010). WRKY genes also play vital roles in the development of plant anthers and embryos (Zhang et al., 2018). GhWRKY33 in upland cotton responds to drought stress and significantly enhances drought resistance when overexpressed in transgenic Arabidopsis (Wei et al., 2020). Overexpression of the rice gene OsWRKY42 reduces cell wall damage caused by high-salinity stress, ultimately enhancing salt tolerance (Pillai et al., 2018). Under low-temperature stress, overexpression of PmWRKY40 gene in plum enhances the cold resistance of plum flowers (Peng et al., 2019).

The first WRKY gene family member, SPF I, was discovered in sweet potato in 1994 and found to be induced by sucrose and PAG (Bu et al., 2020). Subsequently, WRKY genes have been identified in many plant genomes, including Arabidopsis (Eulgem et al., 2000), sweet orange (Silva, Ito & Souza, 2017), sesame (Li et al., 2017), cassava (Wei et al., 2016) and maize (Hu et al., 2021). Recent studies of Rhl have mainly focused on genetic evolution, the development of ornamental aspects, and predictions of MYB gene family functions (Yue, Han & Zhang, 2019; Zhou et al., 2019, 2022; Han et al., 2008). However, there has been no study of the WRKY gene family in this plant. This work used bioinformatics analysis to identify the WRKY genes in Rhl and investigated their expression patterns in different tissues and under abiotic stress conditions. The results lay the foundation to explore the biological functions of WRKY genes in this important sub-species.

Materials and Methods

Prediction of physicochemical properties

The physicochemical properties of the RhlWRKY protein were predicted using the compute PI/MW tool of ExPASy database (http://web.expasy.org/compute_pi/), and subcellular localization was predicted using the WoLFPSORT website (https://www.genscript.com/wolf-psort.html).

Gene identification and phylogenetic tree construction

Firstly, InterProScan (v5.50-84.0) was used to annotate the RhlWRKY protein domain based on InterPro protein database, and the annotation results of pfam gene family were extracted. Other transcription factor families not annotated by InterPro are annotated with PlantTFDB (https://planttfdb.gao-lab.org/). A total of 65 putative WRKY genes were identified, and identify the candidate RhlWRKY from the genome of Rhl (SAR: PRJNA656593) (Zhou et al., 2022). Next, MEGA7.0 was used to conduct cluster analysis of the 90 Arabidopsis protein sequences and the 65 putative RhlWRKY proteins to compare their evolutionary with 1,000 replicates bootstrap analysis for statistical reliability. We further performed a NJ phylogenetic tree of the whole RhlWRKY protein sequences relationships.

Gene structure and conserved motif analysis

A phylogenetic tree was constructed using MEGA7.0, and then the RhlWRKY Protein sequence was used to find conserved motifs by using the Multiple Em for MEME online website (https://meme-suite.org/meme/) with zero and one per sequence, maximum number of motifs sets at 12 and optimum width of motif from 6 to 200. The motif with an e-value less than 1e−10 was retained for further analysis. The genetic structure was determined for the RhlWRKY genes based on the genome Gff3 annotation file. The Gene Structure View (Advanced) software of the TBtools (v2.038) was used to create a phylogenetic tree, motif, and genetic structure diagrams.

Cis-acting elements

The 2,000 bp upstream sequences of the start codons of RhlWRKY genes were collected to obtain the promoter sequences using the GXF Sequences Program of the TBtools. Extract of the and analyzed using the PlantCARE database (Lescot et al., 2002) to predict potential cis-acting elements.

Chromosomal location and collinearity analysis

A chromosomal location map was generated using TBtools and collinearity analysis was performed using McScanX software. The BLAS-BLAST GUI Wrapper algorithm was used for Rhl protein internal self-comparison, the Text Marge for MCscanX for regularizing the gff format file, and the Quick Run MCScanX Wrapper algorithm for self-comparison. Repeat events and collinearity relationships of the RhlWRKY genes were analyzed. The Ka/Ks values for duplicate RhlWRKY gene pairs were calculated using KaKs_Calculator2.0.

Gene ontology functional annotation

Gene Ontology (GO) annotations for each RhlWRKY gene were obtained using Blast2GO software by aligning between the WRKY protein sequences and the NCBI non redundant protein (Conesa et al., 2005). Subsequently, WEGO software (https://wego.genomics.cn/) was used for GO functional classification and to determine gene function distribution (Zhang & Wang, 2005).

Stress treatments and transcriptomic analysis

Seeds of Rhl were collected from Xiaoqinling, Lingbao City, Henan Province. The seeds were decontaminated, sterilized and cultured on MS medium at 25 °C (16 h light/8 h dark). Two-month-old seedlings were subjected to treatments with low-temperature (4 °C), 20% PEG, NaCl (200 mM), and MeJA (200 µM) solutions. Samples were collected at 0, 1, 3, 6, and 12 h with light after treatment and stored at −80 °C. Transcriptome analysis was conducted in May 2023, using material from the hypocotyls of 14-day-old seedlings, as well as roots, stems, leaves, and fully bloomed flowers from three-month-old tissue-cultured plants. Samples were stored at −80 °C for transcriptome sequencing, which was performed in triplicate (three biological and three technical replicates; sequencing services provided by Wuhan Frasergen Bioinformatics Co., Ltd., Wuhan, China). Using SOAPnuke software (v2.1.0) to filter Raw reads, high-quality Clean reads are obtained by removing paired reads containing connectors, reads with a N ratio greater than 0.5%, and reads with a mass value Q ≤ 20 and base number accounting for more than 50% of the entire read. Clean reads were compared with the reference genome using HISAT (v2.2.1) software. The prediction of new transcriptomes was performed using StringTie v1.3.4d software, using the Fragments Per Kilobase of exon model per million mapped fragments values (FPKM values) as an indicator of gene expression level. The distribution of reads on the reference genome generally shows only the distribution of the first 25 longest chromosomes, or Scaold (Fig. S1). DESeq v1.22.2 was used to calculate FPKM ratio (FC, Fold change) and False discovery rate (FDR, False discovery rate) between the difference comparison samples. |log2FC| ≥ 1 and FDR < 0.05 were used as DEGs screening criteria. The statistical power of this experimental design, calculated in RNA SeqPower was shown in Table S1. A transcriptional pattern heatmap of the RhlWRKY gene family was generated using the HeatMap feature in TBtools software (with log2(FPKM) Col Scale settings). The RNA-seq data could be searched from the Biosample with accession numbers SAMN37367964–37367966 (root), SAMN37367967–37367969 (stem), SAMN37367970–37367972 (leaf), SAMN37367973–37367975 (hypocotyl), SAMN37367976–37367978 (flower).

qRT-PCR treatments

RNA was extracted using a magnetic bead total RNA kit from Shanghai Lingjun Biotechnology Co., Ltd., Shanghai, China. The concentration of total RNA and OD260/OD280 values were determined using an ultramicro spectrophotometer (NanoDrop One, Thermo Scientific, Waltham, MA, USA). Electrophoresis was used to identify the integrity of RNA. This RNA was used to synthesize cDNA by reverse transcription. Primers for five genes (see Table S2) were designed using GenScript (https://www.genscript.com) software and synthesized by Sangon Biotech (Shanghai) Co., Ltd., Shanghai, China. The EF1α gene was used as a reference gene. The cDNA from Rhl plants treated with NaCl, PEG, MeJA, and low-temperature treatments was used in qPCR reactions. Relative expression levels were calculated using the 2−ΔΔCt method. Each response had a negative control group and contained three biological and three technical replicates. Actin gene (EF1α) was used as internal reference gene. Results of NTC was not detected. The obtained results were inputted into GraphPadPrism (9.3) software (grouped –mean set-summary data-separated bar graph) to plot expression model diagram, and one-way ANOVA test was used to assess differences by SPSS (13.0) (Analyze–Compare Means-One-way ANOVA), and the P-value of pairwise comparison was obtained by LSD test.

Results and analysis

Characteristics analysis

The results, as shown in Table 1, indicate that the 65 RhlWRKY genes (RhlWRKY_1- RhlWRKY_65) exhibit significant variations in the number of encoded amino acids, ranging from 112 amino acids (RhlWRKY_18) to 1,037 amino acids (RhlWRKY_41). The molecular weight of RhlWRKY proteins ranges from 13.0801 kDa (RhlWRKY_41) to 114.6188 kDa (RhlWRKY_18), and their isoelectric points (pI) range from 4.39 (RhlWRKY_34) to 9.87 (RhlWRKY_41), with an average pI of 6.96. Among these, 26 are alkaline (pI > 7) and 39 are acidic (pI < 7). The hydrophilic coefficients are <0 for all RhlWRKY proteins, indicating all RhlWRKY proteins are hydrophilic proteins. Subcellular localization analysis reveals that the majority of these proteins are predicted to be localized in the cell nucleus, with only a few (RhlWRKY_13, _18, _19, _56, _64, and _65) predicted to be in the chloroplast and cytoplasm. The results revealed that these newly-identified RhlWRKY proteins in Rhododendron henanense exhibited a various subcellular distribution, which may be associated with functional diversification in abiotic stress responses. The similar observations for WRKY proteins had been reported in maize (Hu et al., 2021).

Table 1 Physicochemical properties of RhlWRKY proteins.

Gene ID	Subcellular localization	Number of amino acids	Relative molecular mass	Isoelectric point (PI)	Hydrophilic coefficient	Instability index	Liposoluble index	
RhlWRKY_1	Nucleus	382	41,382.10	6.35	−0.58	48.36	63.90	
RhlWRKY_2	Nucleus	323	35,201.69	6.32	−0.87	68.79	40.77	
RhlWRKY_3	Nucleus	286	32,564.43	4.65	−1.25	77.41	44.62	
RhlWRKY_4	Nucleus	281	30,926.61	7.07	−0.66	45.79	68.36	
RhlWRKY_5	Nucleus	342	38,342.63	9.82	−0.78	49.78	67.54	
RhlWRKY_6	Nucleus	331	36,235.21	9.73	−0.55	40.86	66.31	
RhlWRKY_7	Nucleus	572	62,278.99	8.28	−0.72	61.38	57.50	
RhlWRKY_8	Nucleus	331	36,631.75	5.99	−0.61	54.26	65.68	
RhlWRKY_9	Nucleus	154	16,842.06	9.30	−0.61	40.57	70.26	
RhlWRKY_10	Nucleus	318	34,741.16	9.20	−0.57	45.59	66.23	
RhlWRKY_11	Nucleus	499	54,308.54	5.19	−0.89	56.01	52.99	
RhlWRKY_12	Nucleus	519	56,199.86	5.22	−0.81	45.12	56.88	
RhlWRKY_13	Chloroplast	310	34,423.75	5.15	−0.31	59.25	75.45	
RhlWRKY_14	Nucleus	562	61,577.10	8.58	−0.96	67.04	47.38	
RhlWRKY_15	Nucleus	312	33,955.31	6.51	−0.89	58.99	43.53	
RhlWRKY_16	Nucleus	273	30,816.56	8.74	−0.72	53.25	54.32	
RhlWRKY_17	Nucleus	558	60,808.24	5.36	−0.42	48.65	72.22	
RhlWRKY_18	Chloroplast	1,037	114,618.80	6.21	−0.47	36.20	77.58	
RhlWRKY_19	Chloroplast	235	26,337.87	7.65	−0.93	45.67	46.38	
RhlWRKY_20	Nucleus	513	56,426.35	5.67	−0.91	58.40	60.45	
RhlWRKY_21	Nucleus	327	36,311.32	5.53	−0.80	64.74	54.01	
RhlWRKY_22	Nucleus	706	76,549.48	6.21	−0.77	50.02	58.73	
RhlWRKY_23	Nucleus	314	34,865.89	6.13	−0.68	53.28	60.51	
RhlWRKY_24	Nucleus	530	58,317.74	7.66	−0.82	56.51	56.64	
RhlWRKY_25	Nucleus	353	37,583.77	5.45	−0.55	56.33	55.92	
RhlWRKY_26	Nucleus	548	60,959.07	8.09	−0.96	55.80	46.82	
RhlWRKY_27	Nucleus	341	38,169.53	5.59	−0.74	58.26	48.01	
RhlWRKY_28	Nucleus	368	40,270.70	9.45	−0.51	43.52	72.64	
RhlWRKY_29	Nucleus	382	41,956.37	5.90	−0.74	57.58	58.51	
RhlWRKY_30	Nucleus	335	36,466.57	8.50	−0.82	46.45	64.06	
RhlWRKY_1	Nucleus	669	72,676.20	5.83	−0.79	57.45	53.71	
RhlWRKY_32	Nucleus	337	37,229.15	6.18	−0.71	52.26	65.64	
RhlWRKY_33	Nucleus	265	29,339.31	5.39	−0.86	58.43	50.04	
RhlWRKY_34	Nucleus	321	34,609.14	4.39	−0.57	71.41	64.74	
RhlWRKY_35	Nucleus	324	36,260.44	6.70	−1.12	68.35	41.23	
RhlWRKY_36	Nucleus	529	58,419.62	8.45	−1.00	61.29	42.59	
RhlWRKY_7	Nucleus	333	36,835.36	8.79	−0.70	48.04	67.69	
RhlWRKY_38	Nucleus	539	58,564.21	6.35	−0.68	44.20	61.48	
RhlWRKY_39	Nucleus	596	64,701.54	6.19	−0.72	50.76	60.37	
RhlWRKY_40	Nucleus	333	37,392.63	6.10	−0.95	75.25	45.92	
RhlWRKY41	Nucleus	112	13,080.10	9.87	−0.83	46.54	59.91	
RhlWRKY_42	Nucleus	584	63,627.68	6.88	−0.70	49.00	59.38	
RhlWRKY_43	Nucleus	572	61,829.39	6.28	−0.73	44.64	62.71	
RhlWRKY_44	Nucleus	492	53,384.50	9.07	−0.79	45.97	65.98	
RhlWRKY_45	Nucleus	186	21,067.50	9.44	−0.91	36.35	51.34	
RhlWRKY_46	Nucleus	321	34,644.23	4.39	−0.56	69.21	65.95	
RhlWRKY_47	Nucleus	173	19,797.42	9.64	−0.86	49.91	55.14	
RhlWRKY_48	Nucleus	210	23,878.47	4.54	−0.46	46.85	60.38	
RhlWRKY_49	Nucleus	537	57,845.68	7.27	−0.80	62.56	54.90	
RhlWRKY_50	Nucleus	280	31,264.80	5.44	−0.83	54.87	57.39	
RhlWRKY_51	Nucleus	376	41,384.69	5.85	−0.76	60.19	56.06	
RhlWRKY_52	Nucleus	489	53,513.58	6.40	−1.02	48.52	55.42	
RhlWRKY_53	Nucleus	579	62,838.54	6.77	−0.72	47.18	53.71	
RhlWRKY_54	Nucleus	347	39,043.18	5.46	−0.77	43.47	60.69	
RhlWRKY_55	Nucleus	322	35,377.64	5.67	−0.62	50.87	67.52	
RhlWRKY_56	Nucleus	113	13,129.42	9.76	−0.41	40.76	82.74	
RhlWRKY_57	Nucleus	281	30,786.39	6.71	−0.62	70.46	40.42	
RhlWRKY_58	Nucleus	281	30,929.59	7.07	−0.65	43.96	68.72	
RhlWRKY_59	Nucleus	333	37,447.74	5.08	−0.62	46.62	62.67	
RhlWRKY_60	Nucleus	329	36,314.42	8.14	−0.83	47.06	63.71	
RhlWRKY_61	Nucleus	353	39,742.28	9.74	−0.76	45.65	65.44	
RhlWRKY_2	Nucleus	289	32,291.19	5.23	−0.58	54.88	81.35	
RhlWRKY_63	Nucleus	328	36,623.07	6.43	−0.65	54.85	62.50	
RhlWRKY_64	Cytoplasm	159	18,242.76	9.26	−0.66	22.28	60.06	
RhlWRKY_65	Chloroplast	256	28,908.49	8.19	−0.61	52.76	66.25	

Conservation domain and phylogenetic analysis of the WRKY gene family

A multiple sequence alignment was performed for the conserved domains of the RhlWRKY family, and a SeqLogo was generated (see Fig. S2). The majority of RhlWRKY transcription factors exhibit conservation in the N-terminal heptapeptide domain (WRKYGQK) and the C-terminal zinc finger (C2HH/C). These domains are similar to the amino acid variation patterns observed in WRKY proteins from rice and other species (Zhang & Wang, 2005). In addition to the conserved WRKYGQK sequence, three variants were identified: WKKYGEK in Group I (RhlWRKY36), WRKYGKK in Group II-c (RhlWRKY64), and WRKYGRK in Group II-d (RhlWRKY65).

A phylogenetic tree (see Fig. 1) was constructed by aggregating sequences of WRKY genes from Rhl (65) and Arabidopsis (90). WRKY transcription factors were categorized into three subgroups, with Group II further divided into five subclasses: Group II-a, -b, -c, -d, and -e. The three subgroups are in the following order, from largest to smallest: Group II, Group I, and Group III, with a total of 92, 37, and 27 WRKY genes, respectively.

Figure 1 Phylogenic cluster of WRKY families in R. henanense subsp. lingbaoense. (65), A. thaliana (90).

Gene structure and conserved domains of RhlWRKY

Figure 2 shows that RhlWRKY genes contain 0–9 introns and 1–10 exons, and genes within the same subtribe exhibit similar exon-intron structures. This indicates that the 65 RhlWRKY genes have relatively conserved genetic structures. A total of 12 conserved motifs were identified within the RhlWRKY genes, and are designated as motifs 1–12. Motifs 1, 2, 8, and 4 are found in the majority of Rhl genes, indicating a high level of conservation in these four motifs. Among these, motif1 and motif3 contain the WRKYGQK conserved domain. Motif3, with the presence of CC (cysteine), and motif5, with HH/HC, together form the complete C2HH/C2HC zinc finger. Motif1 (containing C), motif2 (containing C), and motif8, or nearby sequences (containing HH), form a complete C2HH zinc finger. As shown in Fig. 2, nearly all genes contain conserved domains and zinc fingers. In Group I, there are two WRKY domains and two C2H2 zinc fingers. In Group II, there is one WRKY domain and one C2H2 structure, with Group II-a and -b containing a unique motif6 and Group II-d and -e mostly containing motif10, suggesting genes in these groups may have similar functions. In Group III, there is one WRKY domain and one C2HC zinc finger. Within the same subtribe, conserved motifs exhibit similarities, indicating that genes within the same subgroup likely share similar biological functions.

Figure 2 Phylogenetic relationships, conserved motifs and gene structure of R. henanense subsp. lingbaoense.

(A) The phylogenetic tree prepared using sequences of 65 WRKY proteins from R. henanense subsp. lingbaoense. (B) The motif patterns of 65 WRKY Proteins. A total of 12 motifs are shown by the box in different colors. (C) The genetic map is shown. Yellow box, black line, and green box represent CDS, introns, and UTS, respectively.

Analysis of cis-acting elements

A total of 84 elements were identified in the upstream promoter regions of RhlWRKY (Fig. 3), and the results revealed the presence of various cis-elements in the flanking regions associated with stress, hormones, transcription and development etc. The majority of these elements fall into the stress response category, including elements associated with low-temperature, injury, and drought responses. Among the genes, 63 contain low-temperature response elements (MYB), 33 sequences exhibit wound response elements (WUM-motif), and all 65 genes contain drought response elements (MYC). Additionally, 28 genes have TC-rich repeat sequences, known to protect plants from damage in adverse conditions (Sun et al., 2016), and 18 genes feature light response elements (MRE). A total of 37 genes contain W-box elements, participating not only in wound and pathogen stress responses (Jiang et al., 2016) but also in self-regulation and cross-regulation with other genes (Chi et al., 2013). In terms of hormone response elements, 52 genes contain MeJA-induced response elements (CGTCA-motif), and 55 genes have abscisic acid response elements (ABRE). All 65 genes contain transport cis-acting elements (CAAT-box).

Figure 3 The number of various cis acting elements in the WRKY gene of R.henanense subsp. lingbaoense.

Chromosome localization, collinearity analysis and Ka/Ks analysis

The results of chromosomal location analysis (as shown in Fig. 4) indicate that out of the 65 RhlWRKY gene family members, only two genes, RhlWRKY_46 and RhlWRKY_51, were not mapped to definitive chromosomal locations. It may be due to the fact that the scaffold could not be assigned to a chromosome. The remaining RhlWRKY family genes are distributed across 12 chromosomes, with the exception of chromosome 9, indicating a relatively small preference for the chromosomal distribution of RhlWRKY genes. The highest numbers of genes mapped to Chr3, Chr8, and Chr12, each with eight genes, while the fewest genes mapped to Chr1, Chr10, and Chr11, each with three genes. Two genes separated by five or fewer genes and located in a chromosomal segment less than 100 kb in length are considered tandem duplicate genes (Liu et al., 2020). Based on these criteria, a total of eight pairs of tandem duplicate sequences were identified (see Fig. 4, and Table S3), including four pairs on chromosome 5, two pairs on chromosome 8, and one pair on chromosome 6. Collinearity analysis revealed the presence of 97 segmental duplication relationships among the 65 RhlWRKY genes across the 12 chromosomes (see Fig. 5, and Table S3). To study evolutionary factors, Ka/Ks values were calculated for 105 duplicate gene pairs (see Table S3). A Ka/Ks ratio less than 1 indicates that gene pairs are in a state of negative selection or purifying selection, a Ka/Ks ratio greater than 1 indicates positive selection, and a Ka/Ks ratio equal to 1 suggests neutral selection (Wang et al., 2018). The homologous gene pair of RhlWRKY_18 and RhlWRKY_60 has a Ka/Ks ratio greater than 1, indicating positive selection, but the remaining 104 homologous pairs have Ka/Ks values less than 1, suggesting that the vast majority of RhlWRKY genes underwent purification due to environmental pressures.

Figure 4 Chromosomal localization of WRKY Gene in R.henanense subsp. lingbaoense (Note: the red box shows tandem repeat gene pairs).

Figure 5 Collinearity analysis of WRKY genes in R.henanense subsp. lingbaoense.

GO functional annotation analysis

GO annotation was used to classify gene functions into three categories: molecular functions, cellular components, and biological processes. As shown in Fig. 6, WRKY proteins are almost entirely absent in the cellular component category. Instead, these genes are primarily associated with molecular functions related to binding and transcription regulation activity. In terms of biological processes, WRKY proteins are involved in metabolic processes, biological regulation, regulation of biological processes, and cellular processes.

Figure 6 GO functional annotation of RhlWRKY genes in R.henanense subsp. lingbaoense.

Transcriptional abundance analysis of the RhlWRKY gene family

Transcriptional patterns were determined based on RNA-seq data from five different tissues of Rhl: roots, stems, leaves, flowers, and hypocotyls (see Fig. 7). RhlWRKY_56 and _65 were not detected in any of the five tissues, suggesting these might be pseudogenes. There were substantial differences in the expression profiles of various RhlWRKY gene family members across different tissues. Among the 43 RhlWRKY genes, 66.15% were expressed in all five tissues, with 24 of them showing relatively high expression levels (36.92%, FPKM > 4). Combining the RhlWRKY cluster analysis, it is evident that, in Group I, apart from RhlWRKY_7, the rest of the genes exhibited relatively high expression levels in all five tissues, with RhlWRKY_14 exhibiting the highest expression level. In Group II, RhlWRKY_5, _6, _18, _21, _37, _38, _42, _44, and _60 showed relatively high expression levels in all five tissues. RhlWRKY_4, _9, _30, _34, _41, _57, and _64 exhibited lower expression levels or were undetectable in all five tissues. In Group III, RhlWRKY_54, _59, and _62 showed relatively high expression levels in all five tissues, whereas RhlWRKY_17, _25, and _48 had lower expression levels. As shown in Fig. 7, RhlWRKY_14 is highly expressed in all tissues (with FPKM values of 236.62, 390.34, 360.22, 255.24, and 322.50 in roots, stems, leaves, flowers, and hypocotyls, respectively). Additionally, tissue-specific expression patterns were observed, with RhlWRKY_37 showing specificity in stems, leaves, and hypocotyls. Genes with relatively high expression levels in stems included RhlWRKY_6 and _36, and RhlWRKY_60, _42, _21, and _6 exhibited higher expression levels in hypocotyls. Furthermore, RhlWRKY_18 displayed specific expression in flowers, and RhlWRKY_42 and _37 showed tissue-specific expression in roots. These genes likely play key roles in their respective tissues.

Figure 7 Tissue specific expression analysis of RhlWRKY family genes.

The color code shown on the top of the figure represents different log2 values.

Analysis of expression patterns under abiotic stress

Expression analysis was performed on five genes belonging to three subtribes of the WRKY gene family. The five selected genes (RhlWRKY_17,_19,_37,_42,_45) show relatively high expression levels in the selected tissues and were orthologous to Arabidopsis (Fig. 8). Under MeJA treatment, WRKY_17 exhibited significant upregulation at 12 h, and WRKY_37 showed significant down-regulation at 3, 6, and 12 h. WRKY_42 showed significant up-regulation at 1 and 3 h, and WRKY_45 had significant up-regulation at 1 and 12 h with respect to 0 h. Under low-temperature stress, WRKY_17 showed significant up-regulation at 1 and 3 h, WRKY_19 was up-regulated at 3 and 12 h, WRKY_37 was up-regulated at 3 h and further at 12 h, WRKY_42 showed increased expression at 12 h, and WRKY_45 showed higher expression at 3 and 12 h with respect to 0 h. In response to drought stress, WRKY_17 exhibited upregulation at 3 h, WRKY_19 showed significant downregulation at 1 h, and WRKY_37 showed downregulation at 3 and 12 h, with a more significant decrease at 6 h. Additionally, WRKY_42 showed significant upregulation at 1 h, and WRKY_45 had significant upregulation at 3 h and 12 h with respect to 0 h. Under high-salinity stress, WRKY_19 had upregulation at 6 h and 12 h, while WRKY_42 showed a more significant down-regulation at 3 h and 6 h with respect to 0 h. WRKY_17 exhibited a 10-fold upregulation at 12 h under MeJA treatment with respect to 0 h, suggesting this is a major gene that responds to MeJA treatment. WRKY_19 showed an 8.7-fold upregulation at 12 h under low-temperature treatment with respect to 0 h, indicating it as the major gene to respond to low-temperature stress. WRKY_42 showed a 4-fold upregulation at 1 h under drought treatment with respect to 0 h, indicating it as a major gene involved in drought response. Interestingly, WRKY_42 exhibited relatively high expression levels across various tissues (roots, stems, leaves, flowers, and hypocotyls). WRKY_19 showed a 4.5-fold upregulation at 6 h under high-salinity treatment with respect to 0 h, indicating that this gene likely acts to counter high-salinity stress.

Figure 8 Expression of 5 WRKY genes under different abiotic factor stress in R. henanense subsp. lingbaoense.

(A) MeJA (B) 4 °C (C) PEG (D) NaCl. Statistical analysis is performed using an one-way analysis of variance (Note *P < 0.05 **P < 0.01 ***P < 0.001).

Discussion

WRKY transcription factors activate or inhibit the expression of downstream genes through self-regulation or cross-regulation, participating in various biological processes of plant growth, development, and stress responses (Sun et al., 2020). The WRKY gene family has been studied in various plants such as Arabidopsis (Eulgem et al., 2000), sweet orange (Silva, Ito & Souza, 2017), sesame (Li et al., 2017), cassav ((Wei et al., 2016), maize (Hu et al., 2021), sunflower (Li et al., 2020), and osmanthus (Ding et al., 2020). However, this identification and characterization of WRKY transcription factors in Rhl represents a novel contribution to the field. In this study, a total of 65 RhlWRKY genes were identified, and subcellular localization analysis revealed that most RhlWRKY proteins are predicted to be located in the cell nucleus, consistent with the fundamental characteristic of transcription factors being nuclear proteins (Fu et al., 2019) (Table 1). Based on phylogenetic analysis (Fig. 2), the RhlWRKY gene family can be divided into three subtribes, Group II, I, and III, in order from largest to smallest. The distribution is similar to that in Arabidopsis. Ten pairs of homologous genes were found between Arabidopsis and Rhl, suggesting these genes may have similar functions (Zhu, 2021). For instance, overexpression of Arabidopsis AtWRKY75 accelerates leaf aging (Zhang et al., 2021), and overexpression of its orthologous gene RhlWRKY_45 may also accelerate leaf aging. Multiple sequence alignment (Fig. S2) showed that most RhlWRKY proteins share a conserved heptapeptide sequence (WRKYGQK) at their N-terminus and a shared zinc-finger at their C-terminus. However, other variants, such as WKKY-GEK, WRKYGKK, and WRKYGRK, were also found in this analysis. This variation has been observed for other WRKY genes in Arabidopsis, rice, apple, grape, and potato (Eulgem et al., 2000; Ross, Yue & Shen, 2007; Meng et al., 2016; Guo et al., 2014; Zhang et al., 2017). Perhaps these different variants will bind to distinct elements, not just W-box elements (Eulgem et al., 2000). Eight WRKY proteins lacking a complete zinc-finger motif and four proteins with incomplete WRKYGQK sequences were also identified as members of the RhlWRKY gene family. Gain and loss of structural domains may partially explain the expansion of the WRKY gene family in Rhl.

Conserved motif analysis (see Fig. 2) revealed that the majority of WRKY genes possess motif1, motif2, motif8, and motif4, suggesting these may be important features for identifying WRKY genes. Similar conserved motifs exist within the same subtribe, but significant differences can be observed between different subgroups, consistent with the diverse functions of the WRKY gene family (Zheng et al., 2022). Genetic structure analysis (Fig. 2) showed that homologous genes have a generally similar structure, but there are also irregularities. For example, the two genes in the homologous gene pair of RhlWRKY_17 and RhlWRKY_25 vary in the number of introns, with four and two, respectively. This discrepancy might be due to the loss, gain, or alteration of introns during the evolution of the WRKY gene family (Rogozin et al., 2005). Genes with intron loss can accelerate evolution through duplication processes (Lecharny et al., 2003).

The analysis of cis-acting elements (Fig. 4) revealed that most are related to stress responses, including elements responsive to MeJA, low-temperature, and drought. Chromosomal mapping (Fig. 4) indicates that 28.6% of RhlWRKY genes are clustered, which can facilitate sequence exchange between genes (Shui et al., 2023). As shown in Fig. 4, there are eight pairs originating from tandem duplication events, and 97 pairs of segmental duplication events were found to have a collinearity relationship. This suggests that segmental duplication played a crucial role in the expansion of this family (Li et al., 2022). Previous research has shown that gene duplication largely explains the emergence of new genes (Yin et al., 2013). Gene duplication can lead to sub-functionalization. For example, the function of wheat WRKY gene family members expanded through tandem and whole-genome duplications (Hassan et al., 2019). Analyzing the Ka and Ks substitution rates in duplicate genes provides insight into the evolution of important genes (Hanada, Shiu & Li, 2007). In 104 out of 105 duplicated pairs, the Ka/Ks ratio is <1 (see Table S3), indicating that these gene pairs have undergone strong purifying selection. Purifying selection typically removes harmful alleles selectively over time (Biswas & Akey, 2006), suggesting that the WRKY gene family likely plays a significant role in the development and survival of Rhl, making the preservation and propagation of all its members necessary. Only one pair of homologous genes, RhlWRKY_18 and RhlWRKY_60, has a Ka/Ks ratio >1, indicating positive selection and carrying important implications for species evolution (Li et al., 2023). According to the results of GO functional annotation (see Fig. 6), RhlWRKY transcription factors perform various molecular functions and regulate various cellular metabolic processes.

Transcriptome expression profile analysis reveals that three genes, RhlWRKY_4, _56, and _65, show minimal expression levels in the roots, stems, leaves, flowers, and hypocotyls of Rhl. In five tissues, 24 RhlWRKY members exhibit relatively high expression levels in these five tissues, while several of the remaining RhlWRKYs exhibit significant tissue-specific expression patterns. This suggests that RhlWRKYs may have various roles in regulating growth, development, and secondary metabolism in Rhl. Interestingly, RhlWRKY_8, _23, and _32 have higher expression levels in flowers. These three genes have orthologous genes in Catharanthus roseus that regulate the biosynthesis of terpenoid indole alkaloids, such as CrWRKY1 (At3G56400 and At2G40750 orthologous genes) (Schluttenhofer et al., 2014). Therefore, RhlWRKY_8, _23, and _32 may be involved in the regulation of alkaloid secondary metabolism in Rhl. The essential roles of WRKY TFs in plant growth, development, and stress resistance have been extensively studied, particularly in Arabidopsis, where many WRKY genes have been functionally characterized. Thus, identifying the closest Arabidopsis homologs for individual RhlWRKY genes can provide hints about their potential functions. For example, At5G07100 regulates heat shock proteins and heat-induced ethylene-dependent responses (Li, Fu & Chen, 2011), so its orthologous gene RhlWRKY_26 may have a similar function. Both At4G23810 and At3G56400, which belong to Group III, play important roles in leaf aging (Chen et al., 2010) RhlWRKY_54 and _59 also belong to Group III and are highly expressed in leaves. The orthologous counterpart of RhlWRKY_15 is AT1G69310 in Arabidopsis, which can enhance drought tolerance by increasing abscisic acid levels (Srivastava et al., 2018). AT4G31800 and AT2G25000 enhance plant sensitivity to salt and osmotic stress, suggesting potential function for the orthologous RhlWRKY_60 (Chen et al., 2010). RhlWRKY_60 may also have a similar function, given its high expression levels in stems and hypocotyls. AT4G31550 and AT2G24570 act to defend against both biotic and abiotic stress (Liu et al., 2011). Their orthologous gene, RhlWRKY_10, may have similar functions, especially given its tissue-specific expression in roots.

The expression analysis (Fig. 8) indicates that WRKY_17 is the major gene whose expression changes in response to MeJA treatment, WRKY_19 is the major gene with changes in response to low-temperature and high-salinity treatments, and WRKY_42 is the major gene that changes in response to drought treatment. RhlWRKY_37 is an orthologous gene to Arabidopsis AtWRKY40, which is significantly expressed in response to drought stress, indicating a drought resistance function (Che et al., 2018). Interestingly, as shown in Fig. 8, the expression of RhlWRKY_37 is significantly downregulated after 3 h of drought treatment, which is opposite to the behavior of the AtWRKY40 gene. This difference might be due to variations in the environmental preferences of the two plants, as Rhl prefers partial shade to intense sunlight (Han et al., 2008), whereas Arabidopsis typically grows in dry outdoor soils (Zhao, 2011). Additionally, RhlWRKY_37 has very high expression levels in roots, stems, leaves, flowers, and hypocotyls. AtWRKY6 and RhlWRKY_42 are a homologous gene pair, and AtWRKY6 regulates seed germination. When its expression is low, it reduces the sensitivity of seeds to ABA, leading to faster seed germination, and with higher expression, germination is slower (Li et al., 2017). Thus, in addition to response to drought stress, the RhlWRKY_42 gene may also regulate seed germination. RhlWRKY_42 also exhibits high expression in roots, stems, leaves, flowers, and hypocotyls, making it a candidate gene for further research on the function of RhlWRKY genes.

Conclusion

This study identified 65 WRKY genes in the genome of Rhl and analyzed their gene characteristics and expression patterns in different tissues and under various stress conditions. The results showed that RhlWRKY genes can respond to various hormones and abiotic stresses. Specifically, WRKY_42 and WRKY_17 were identified as genes changing in response to drought and MeJA treatment, respectively, and WRKY_19 showed changes in expression in response to low-temperature and high-salinity conditions. These findings lay the foundation for further work to identify and characterize WRKY genes in Rhododendron. Additionally, through phylogenetic comparisons, the study made functional predictions for some genes and identified valuable candidate genes for RhlbZIP gene molecular mechanism, providing a basis for further functional research and feature analysis. This study laid a solid foundation for further analysis of, function and characteristics.

Supplemental Information

Supplemental Information 1 Sample randomness distribution curve.

Supplemental Information 2 Conserved domain SeqLogo of the WRKY family of R.henanense subsp.lingbaoense.

Supplemental Information 3 Supplemental Tables.

Values of Ks, Ka, and Ka/Ks for Duplicate Gene Pairs

Supplemental Information 4 WRKY genes expression raw data.

Supplemental Information 5 MIQE checklist and detailed answer.

Supplemental Information 6 Raw data exported from the CFX96 Real time PCR Detection System (Bio Rad, USA) for data analyses and preparation for Fig. 8 for the different stress treatments time of 0,1,3,6,12h.

Supplemental Information 7 Raw data exported from the DNBSEQ-T7 (Wuhan Feisha Bioinformation Co., Ltd) for data analyses and preparation for Fig. 7 and Table S1 for the shoot, root, leaf, flower, and hypocotyl of Rhl.

Additional Information and Declarations

Competing Interests

Author Contributions

Data Availability

The authors declare that they have no competing interests.

Xiangmeng Guo conceived and designed the experiments, performed the experiments, authored or reviewed drafts of the article, and approved the final draft.

Xinyu Yan performed the experiments, analyzed the data, prepared figures and/or tables, and approved the final draft.

Yonghui Li conceived and designed the experiments, performed the experiments, analyzed the data, prepared figures and/or tables, authored or reviewed drafts of the article, and approved the final draft.

The following information was supplied regarding data availability:

The RNA-seq data is available at NCBI: SAMN37367964–37367966 (root); SAMN37367967–37367969 (stem); SAMN37367970–37367972 (leaf); SAMN37367973–37367975 (hypocotyl); SAMN37367976–37367978 (flower).

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
