# Peer review of "Genome-wide identification and expression analysis of the WRKY gene family in Rhododendron henanense subsp. lingbaoense"

_PeerJ, doi:10.7717/peerj.17435_

## Round 0.1 · original submission · Major Revisions

The authors are advised to revise the manuscript as per reviewers suggestions.

Reviewer 1 ·

Basic reporting

no comment

Experimental design

- In lines 88, please elaborate the criteria that you used to identify the 65 putative WRKY genes.
- In lines 88 and 89, please describe what specific methods or algorithms were used in the cluster analysis.
- In line 100, please describe what specific methods or algorithms were used in the collinearity analysis.
- In line 170-171, please describe the specific criteria that were used to classify the 84 elements into four functional categories in the method section.
- Please describe what statistical analyses were used to compare expressions across the 5 genes in the method section. You mentioned that ANOVA test was used for the comparison. But ANOVA only generates a general p-value to show if there is a significant difference between one of the pairwise comparisons among all the comparisons across the 5 genes. What test did you use to get the pairwise comparison p-values?

Validity of the findings

no comment

Reviewer 2 ·

Basic reporting

Guo et al. described the attributes of the WRKY transcription factor family in Rhl through a comprehensive integration of multiple methodologies. They also conducted a comparative analysis of gene expression variations across five tissue types and under various conditions, shedding light on the potential roles of RhlWRKY genes in plant development. However, a thorough reevaluation of the data is imperative before acceptance.

Experimental design

1. The bioinformatic analysis methodology requires greater elucidation. It is recommended that authors furnish comprehensive details on all parameters utilized, along with the specific software employed. Additionally, clarification regarding the utilization of default parameters needs to be announced as well.

2. More extensive details of the RNA-seq analysis procedures are needed, encompassing quality control measures, read trimming, genetic alignment protocols, and criteria employed for differential expression analysis (e.g., fold change and cutoff values).

3. Concerning the identification of 65 putative WRKY genes in Rhl, the BlastP-based approach failed to pinpoint Rhl-species-specific WRKY genes, which were only present in Rhl but absent in Arabidopsis. Identification of orthologous genes between two species should be meticulously executed using alternative software, such as OrthoFinder. Hence, it is recommended to reanalyze gene identification procedures and reconstruct the phylogenetic tree.

Validity of the findings

1. The sequencing depth of individual samples is pivotal for accurate comparisons of gene expression across distinct conditions. It is advised to present the sequencing depth of the RNA-seq samples graphically.

2. A comprehensive account of the statistical analysis should be incorporated into the Materials and Methods section, particularly elucidating the methodologies employed for analyzing expression patterns under abiotic stress conditions.

Reviewer 3 ·

Basic reporting

The present study identifies WRKY gene family members in Rhododendron henanense subsp. lingbaoense. Using in silico analysis, the authors describes 65 WRKY genes in this plant species. Further, various features of the WRKY genes such as their phylogeny, gene expression pattern and cis acting elements have been studied. The expression of three WRKY genes have been shown to be responsive to the abiotic stresses. In general, the manuscript is well-written with requisite illustrations and tables. The results of the present study should be useful for the audience working in the area of regulatory roles of WRKY genes.

Experimental design

The experiments are well-designed and in line with the objectives of the present study.

Validity of the findings

The conclusions are well stated.

Additional comments

• Line 27: Indicating purification should be replaced with indicating “purifying selection”.

• The conclusion section (line 32-35) of the abstract should be rewritten. The authors have mentioned “change in response” of the gene, which may be replaced by modulation in the expression of these genes or something similar.
• The plant species name should be checked properly. It should be italicized.

• The database of the Rhl, used for the identification of WRKY gene should be clearly mentioned.

• What were the light conditions during the stress treatment?

• Certain WRKY genes were predicted to be localized in chloroplast. How the authors will explain this observation? Whether the similar observations for WRKY proteins have been reported in other plants?
• Cis acting elements>>> cis should be italicized.
• Line 184: gene disassembly? It may be due to the fact that the scaffold could not be assigned to a chromosome.
• While describing up or downregulation, the reference should be mentioned i.e upregulation/downregulation with respect to what.

---

## Round 0.2 · accepted · Accept

The authors have revised the manuscript thoroughly and it may be accepted.

Reviewer 1 ·

Basic reporting

The authors have sufficiently addressed my comments.

Experimental design

The authors have sufficiently addressed my comments.

Validity of the findings

The authors have sufficiently addressed my comments.

Reviewer 2 ·

Basic reporting

NA

Experimental design

NA

Validity of the findings

NA

Additional comments

The authors addressed all of my questions satisfactorily. I recommend accepting!

Reviewer 3 ·

Basic reporting

The authors have addressed my queries in the revised manuscript.

Experimental design

OK

Validity of the findings

Ok